# Calprotectin as a Biomarker for Infectious Diseases: A Comparative Review with Conventional Inflammatory Markers

**DOI:** 10.3390/ijms26136476

**Published:** 2025-07-04

**Authors:** Kristina Sejersen, Mats B. Eriksson, Anders O. Larsson

**Affiliations:** 1Department of Medical Sciences, Uppsala University, Uppsala University Hospital, 75185 Uppsala, Sweden; kristina.sejersen@unilabs.com (K.S.); anders.larsson@akademiska.se (A.O.L.); 2Unilabs AB, 17154 Stockholm, Sweden; 3Department of Surgical Sciences, Uppsala University, Uppsala University Hospital, 75185 Uppsala, Sweden; 4NOVA Medical School, New University of Lisbon, 1099-085 Lisbon, Portugal

**Keywords:** bacterial, biomarker, calprotectin, infection, inflammation, neutrophil

## Abstract

Calprotectin, the most abundant cytosolic protein in neutrophils, is a S100A8/S100A9 heterodimer released during immune activation. It inhibits bacterial growth by binding to essential metal ions and contributes to inflammation and leukocyte migration. This review highlights calprotectin’s potential as a diagnostic marker for bacterial infections and inflammation. Clinical trials demonstrate that calprotectin is at least as effective as C-reactive protein, procalcitonin, and white blood cell counts in predicting bacterial infections. The rapid elevation of calprotectin levels in the early stages of sepsis, pneumonia, brain injury, and transplant complications underscores its diagnostic value. Predictive use of calprotectin may reduce ICU stays, mortality, and costs. However, challenges remain, including assay standardization and bacterial–viral differentiation. Advanced methods, such as the particle-enhanced turbidimetric immunoassay, enable faster and more reliable measurements. While calprotectin shows promise, further standardization and clinical validation are necessary to optimize its diagnostic utility.

## 1. Introduction

Calprotectin was first identified in 1980 as an antimicrobial protein located in the cytosol of neutrophil granulocytes [1]. It was later recognized as a promising biomarker for inflammation [2,3]. Research has also shown that calprotectin contributes to the recruitment of inflammatory cells by interacting with endothelial cells (ECs) [4], and that it may influence physiological homeostasis through its zinc-binding properties [5].

Initially referred to as major leukocyte protein L1 or 27E10, calprotectin was subsequently characterized as a heterodimer consisting of the proteins S100A8 and S100A9. These subunits have been known by several other names, including myeloid-related proteins 8 and 9 (MRP-8 and MRP-9), migration-inhibitory-factor-related proteins of 8 and 14 kDa (MRP-8 and MRP-14), calgranulin A and B, cystic fibrosis antigen, and alarmins [6,7].

The diverse functions of calprotectin are primarily linked to active inflammatory processes. These include antimicrobial defense mechanisms and T helper cell type 1 (Th1)-mediated immune responses, such as those seen in allograft rejection or autoimmune diseases.

### Aims of the Review

The overarching goal of this review is to assess the potential of calprotectin measurement in blood and other body fluids to enhance the diagnosis of infection and inflammation. A key objective is to investigate how elevated calprotectin levels correlate with the presence of infections—particularly bacterial infections—as well as their relationship to various inflammatory conditions.

## 2. Bacterial Infections—An Overview

Bacterial infections are caused by pathogenic bacteria that either proliferate within the body or release toxins harmful to host cells. They are a major global health concern and are responsible for a wide range of diseases, including pneumonia, tuberculosis, meningitis, osteomyelitis, gastroenteritis, cholera, periodontitis, conjunctivitis, and sepsis [8,9,10,11,12,13,14,15,16,17].

The immune system defends the body through a complex network of biological processes that recognize and eliminate invading pathogens to maintain homeostasis. The innate immune system employs various pattern recognition receptors (PRRs) to detect pathogen-associated and damage-associated molecular patterns (PAMPs and DAMPs), initiating rapid host defense mechanisms [18]. For instance, DNA sensors such as cyclic GMP-AMP synthase (cGAS) detect cytosolic DNA and produce the second messenger 2′-3′-cyclic GMP-AMP (2′3′-cGAMP), which binds to the stimulator of interferon genes (STING). This interaction triggers downstream signaling from the endoplasmic reticulum to the Golgi apparatus, leading to activation of TANK-binding kinase 1 (TBK1) and interferon regulatory factor 3 (IRF3), ultimately promoting type I interferon (IFN-I) and inflammatory cytokine production [19,20,21,22].

Despite advancements in healthcare, infectious diseases remain a leading cause of mortality, accounting for more than 10 million deaths annually. A study published in The Lancet estimated that sepsis alone was responsible for approximately 11 million deaths worldwide in 2017 [23]. Sepsis continues to be a critical global health issue, prompting the World Health Organization (WHO) to designate it a global health priority [24]. It remains one of the most common reasons for ICU admission and is a major contributor to patient mortality [25,26,27].

Although the discovery of antibiotics revolutionized medicine and has saved countless lives [28], their overuse and misuse have led to the emergence of multidrug-resistant (MDR) bacteria [29]. This poses a growing global threat, with projections estimating that antimicrobial resistance (AMR) will directly cause 1.91 million deaths and contribute to an additional 8.22 million deaths annually by 2050 [30].

Early and accurate diagnosis is critical to preventing complications, such as sepsis, and to minimizing mortality, morbidity, and unnecessary antibiotic use. Traditionally, diagnosis is based on clinical symptoms supported by biomarkers like white blood cell (WBC) count and C-reactive protein (CRP). However, these tools fail to identify bacterial infections in approximately 40% of cases [31].

The gold standard for confirming a bacterial infection remains microbial culture, often from blood samples. These cultures typically require at least 24 h for results and are positive in only about 50% of sepsis cases [32]. Culture sensitivity is limited by prior antibiotic treatment, sampling errors, and the presence of slow-growing or fastidious organisms. Moreover, sepsis may also be caused by viral, fungal, or parasitic infections not detectable by standard cultures [33].

Sepsis often presents with non-specific symptoms, making diagnosis challenging [34,35]. Any delay in treatment significantly increases mortality risk [36,37]. The 2016 Sepsis-3 consensus redefined sepsis as life-threatening organ dysfunction resulting from a dysregulated host response to infection. This revised definition replaced the earlier systemic inflammatory response syndrome (SIRS) criteria and introduced the Sequential Organ Failure Assessment (SOFA) score as a clinical tool to assess disease severity [38]. While the SOFA score helps predict outcomes, especially in the early stages of disease, it does not identify patients before they require ICU care and is not specific to the underlying cause [38,39].

Given the limitations of culture-based methods for early diagnosis, biomarkers are increasingly used to distinguish bacterial infections from viral or fungal ones. Common biomarkers include WBC count; neutrophil count; CRP; procalcitonin (PCT); and, to a lesser extent, calprotectin [40,41].

The inflammatory response varies depending on the type of pathogen—bacterial, viral, or parasitic—with distinct mediators and target tissues involved [42]. A well-orchestrated acute inflammatory response results in pathogen clearance followed by tissue repair and resolution, driven in part by resident macrophages [43].

## 3. Inflammation and Chronic Inflammatory States

Inflammation is a fundamental response of the human immune system to harmful stimuli. However, when inflammation becomes chronic, it can lead to various secondary effects in the biological response that are linked to an elevated risk of developing chronic diseases and disorders. Chronic inflammation typically occurs in the absence of an ongoing external stimulus. This can result from unresolved infections that evade endogenous defense mechanisms or resist immune clearance. Other causes include exposure to physical or chemical agents that cannot be effectively degraded, as well as genetic predispositions. Persistent foreign bodies, ongoing chemical exposure, repeated episodes of acute inflammation, and specific pathogens are all recognized as key contributors to chronic inflammation [44,45].

## 4. Physicochemical Properties and Pathophysiological Data of Calprotectin

Calprotectin is a calcium- and zinc-binding protein belonging to the S100 protein family. The structure of calprotectin is displayed in Figure 1. It is predominantly expressed in myelomonocytic cells—primarily neutrophils and monocytes, and to a lesser extent in immature macrophages [7,46]. Calprotectin constitutes approximately 40–60% of the cytosolic proteins in neutrophils and around 5% in monocytes/macrophages. It is rapidly released upon activation of neutrophils [7,47,48].

Structurally, calprotectin is a heterodimer made up of S100A8 and S100A9 proteins. Each monomer features a helix-loop-helix motif and is capable of binding two Ca^2+^ ions, as well as other divalent metal ions like Zn^2+^. These ion-binding properties influence oligomerization and functional activity [50,51,52,53]. While S100A8 and S100A9 can form homodimers, heterodimers, or heterotetramers, the heterodimer is the most stable form and is responsible for most of calprotectin’s biological effects [54]. These proteins are expressed separately, but S100A8 exhibits a faster turnover. In the absence of S100A9, as observed in knockout mice, S100A8 serum levels are almost undetectable [55]. Besides, bacteria are typically recognized by Toll-like receptors (TLRs) on tissue-resident macrophages, leading to the release of pro-inflammatory cytokines such as tumor necrosis factor-alpha (TNF-α), interleukin-1 beta (IL-1β), and interleukin-6 (IL-6), as well as chemokines like CCL2 (MCP-1) and CXCL8 (IL-8), and lipid mediators such as prostaglandin E2 (PGE2) [55]

Although classically expressed in granulocytes, monocytes, and early-stage macrophages after activation by PAMPs or DAMPs, S100A8 and S100A9 can also be expressed by other cell types—such as endothelial cells, keratinocytes, osteoclasts, chondrocytes, and fibroblast-like synoviocytes—under certain conditions [56,57,58,59,60]. During inflammation, calprotectin is released at the site, with levels increasing by more than 100-fold, making it a powerful acute-phase reactant [61,62].

Functionally, calprotectin is involved in cytoskeletal regulation, leukocyte migration and trafficking, and inflammatory amplification—key aspects of host defense against infections. It also exhibits antimicrobial activity by chelating essential metal ions like Mn^2+^ and Zn^2+^, thereby depriving microbes of nutrients [63].

Calprotectin participates in both intracellular and extracellular immune functions. Intracellularly, it influences immune cell signaling pathways and modulates inflammatory responses, including leukocyte chemotaxis and tissue infiltration. Extracellularly, it binds to receptors such as RAGE and TLR4, triggering signaling cascades via MyD88 and NF-κB, which lead to the production of proinflammatory cytokines like TNF-α, IL-6, IL-8, and IL-23 [61].

Calprotectin also plays a key role in the adaptive immune response by promoting CD8^+^ T cell induction during antigen presentation [64]. It acts as a co-stimulatory molecule in combination with CD40/CD40L signaling, contributing to the breakdown of T cell tolerance. In murine models lacking S100A8/A9, reduced IL-17 production and lower levels of autoantibodies were observed, implicating calprotectin in autoimmune pathogenesis [64].

### 4.1. Calprotectin’s Role in Infection Diagnosis

Circulating calprotectin, which is detectable in plasma or serum, is a valuable biomarker for diagnosing infections. A schematic illustration highlighting some key features of calprotectin are displayed in Figure 2. It is particularly useful for identifying and evaluating the severity of bacterial infections and sepsis. It is released promptly into the bloodstream following neutrophil activation during infection, enabling early identification of the body’s inflammatory response, often before notable increases in conventional markers like C-reactive protein (CRP) and procalcitonin (PCT) occur [65,66,67]. Recent research indicates that calprotectin levels can rise within two hours of inflammation onset [68].

Circulating calprotectin concentrations are markedly higher in bacterial infections than in viral or non-infectious conditions. This makes calprotectin a helpful marker for distinguishing bacterial from viral causes, especially in cases of acute respiratory infections and sepsis. Multiple studies have shown that calprotectin surpasses or complements other biomarkers, such as PCT and HBP, in distinguishing between bacterial and viral infections. Furthermore, evidence suggests that calprotectin is more effective than PCT and HBP in distinguishing bacterial, mycoplasma, and viral infections.

Beyond its diagnostic role, circulating calprotectin is closely associated with infection severity and prognosis. Elevated levels are commonly observed in patients with sepsis and multi-organ dysfunction. Higher concentrations at admission are associated with an increased risk of adverse outcomes, including 30-day mortality in critically ill and septic patients [65,66,67,69,70]. The relationship between calprotectin levels and infection severity highlights its usefulness in risk assessment and guiding clinical decisions, particularly in emergency and intensive care settings.

Furthermore, circulating calprotectin has shown utility in the context of SARS-CoV-2 infection, with significantly higher levels observed in patients with confirmed infection compared to controls, supporting its role in early diagnosis and patient management [71].

However, calprotectin is not exclusively specific to infections; it is also a sensitive indicator of neutrophil-mediated inflammation and may be elevated in other inflammatory disorders, such as autoimmune diseases. Therefore, its levels should always be interpreted within the broader clinical context [72].

Calprotectin is increasingly recognized as a valuable marker of infection and inflammation in various clinical specimens, including body fluids such as synovial fluid, cerebrospinal fluid (CSF), and pleural fluid, beyond blood and stool.

Urine calprotectin has shown promise as a useful biomarker for detecting urinary tract infections (UTIs). Studies have shown that people with UTIs have significantly higher levels of urinary calprotectin. Its diagnostic performance is comparable to or better than conventional indicators, such as dipstick pyuria. Importantly, calprotectin can detect infections when pyuria is absent, which could improve diagnostic sensitivity in atypical cases [73].

Calprotectin in synovial fluid is a highly sensitive and specific biomarker for diagnosing periprosthetic joint infection (PJI). Meta-analyses and clinical studies report sensitivity and specificity rates exceeding 90%, establishing synovial calprotectin as a reliable, rapid diagnostic tool for distinguishing septic from aseptic joint failure, even in complex clinical scenarios. Calprotectin can be measured using ELISA or lateral flow assays. Its low cost and high diagnostic accuracy support its use in routine clinical practice [74,75,76].

Calprotectin concentrations in CSF are notably increased in individuals with infections or inflammatory disorders of the central nervous system. Research indicates that measuring CSF calprotectin levels is useful for distinguishing bacterial meningitis from non-bacterial forms, demonstrating strong sensitivity and specificity. Elevated calprotectin levels also signal neuroinflammatory processes and are significantly higher in patients with neurological infections than in healthy controls. However, while calprotectin is a reliable indicator of inflammation, it does not always allow for clear differentiation between infectious and non-infectious inflammatory conditions within the CNS [77,78].

Calprotectin concentrations in pleural fluid are significantly higher in bacterial infections, such as parapneumonic effusions and empyema, than in malignant pleural effusions. Research indicates that benign pleural effusions, particularly those caused by infections such as pneumonia or tuberculosis, often exhibit calprotectin levels in the thousands of ng/mL. For instance, one study reported median calprotectin concentrations of 3517.9 ng/mL in effusions related to pneumonia and 2982.3 ng/mL in effusions due to tuberculosis. In contrast, malignant effusions had a median concentration of approximately 257.2 ng/mL [79].

Measuring calprotectin in synovial fluid, CSF, and pleural fluid provides valuable clinical information for diagnosing infections and inflammatory conditions in these compartments. Calprotectin’s high sensitivity and specificity, rapid turnaround time, and ease of measurement make it a promising tool for clinicians in diverse diagnostic settings.

### 4.2. Method for Calprotectin Analysis: Particle-Enhanced Turbidimetric Immunoassay (PETIA)

Enzyme-linked immunosorbent assay (ELISA) has traditionally been the most widely employed method for quantifying calprotectin levels. However, this technique is often limited by prolonged turnaround times. In contrast, the more recently developed particle-enhanced turbidimetric immunoassay (PETIA) offers a significantly faster alternative. PETIA is generally implemented as a random-access assay, meaning samples are processed on arrival in the laboratory, which contributes to shortened analytical turnaround times.

ELISAs typically incur higher costs compared to turbidimetric assays, largely due to increased labor requirements [80]. ELISA kits are available with both monoclonal and polyclonal antibodies, while chemiluminescent immunoassays (CLIA) employ monoclonal antibodies for fecal calprotectin detection.

The GCAL assay, a PETIA method utilizing polyclonal antibodies, is marketed as an open-channel assay and is compatible with a variety of general-purpose clinical chemistry analyzers. Another PETIA approach, using immunoglobulin Y (IgY) antibodies derived from egg yolk, has been validated for calprotectin quantification in plasma and serum samples [81]. Avian antibodies offer the advantage of minimal interference from common analytical disruptors such as rheumatoid factor (RF), human anti-mouse antibodies (HAMA), or the human complement system—known issues with mammalian-derived antibodies [82,83].

The PETIA method used in this study is compatible with a broad range of clinical chemistry analyzers and allows for rapid analysis, delivering results in as little as 10 min.

## 5. White Blood Cell (WBC) Count

White blood cells (WBCs) are essential components of the immune system and are involved in a wide range of pathological conditions, including infections, inflammatory diseases, and cancer. WBCs are broadly classified into three subpopulations: lymphocytes (T and B cells), monocytes, and granulocytes (neutrophils, eosinophils, and basophils) [84,85,86].

The WBC count is a widely utilized, cost-effective, and straightforward biomarker of systemic inflammation. As primary effectors of innate immunity, WBCs are crucial for orchestrating immune cell interactions [87,88]. During infections, these cells are rapidly recruited to affected tissues [89]. Their lifespan in circulation and tissues is approximately 24 h, after which they undergo apoptosis and are cleared by macrophages [87,88,89,90].

WBCs are distributed throughout the body and are produced in the bone marrow from multipotent hematopoietic stem cells [91]. Leucocytosis, or elevated WBC count, typically refers to values exceeding 11 × 10^9^ cells/L in adults, although normal reference ranges vary by age [92,93]. While leucocytosis is a hallmark of acute inflammation, it remains a non-specific indicator [94,95].

Congenital leukocyte disorders are rare, whereas acquired leukocyte abnormalities are more common in developed countries [96]. Bone marrow suppression may lead to leucopenia, often observed in hematological diseases. Other contributing factors include nutritional deficiencies (e.g., vitamin B12, folate, copper), toxins, radiation, autoimmune conditions, and infections such as HIV and SARS-CoV-2 [96,97,98]. Numerous drugs can also cause leucopenia by impairing myeloid progenitors or inducing apoptosis [99], with chemotherapy being a major cause in high-income regions [100,101].

Neutrophils constitute 50–70% of circulating WBCs [102]; thus, leucopenia is frequently synonymous with neutropenia [87].

Chronic inflammation plays a pivotal role in atherosclerosis, as well as in elevated WBC counts.

Subtypes such as neutrophils, monocytes, and eosinophils also demonstrate such associations [103,104]. Monocytes that infiltrate vascular tissues differentiate into macrophages; contribute to lipid uptake; and together with neutrophils form the inflammatory plaque, increasing cardiovascular risk [105].

In three hundred fifty-one patients presenting to an emergency department who triggered the sepsis alert, 26% were immediately transferred either the intensive care unit (ICU) or high-dependency unit (HDU). Elevated levels of calprotectin were significantly associated with such transfer. The best plasma calprotectin cut-off, 4.0 mg/L, showed a sensitivity of 42.5% and specificity of 83% for transfer to the ICU/HDU among patients with infection and superior to traditional biomarkers in predicting the need for transfer to the ICU/HDU [106]. In 110 ICU patients’ plasma, calprotectin was evaluated as an early marker of bacterial infections and compared with PCT, CRP, and WBC. In the 58 patients who developed a suspected or confirmed bacterial infection, plasma calprotectin predicted such infections significantly better than what WBC and PCT did, and marginally better than what CRP did [65]. In a study on the health and economic implications of calprotectin as a predictive tool to initiate antimicrobial therapy in a cohort of critically ill patients, this protein was evaluated under the assumption that calprotectin was used predictively and comparators (white blood cells, procalcitonin, and C-reactive protein) were used diagnostically. If calprotectin would be used predictively, it was hypothesized that cost-effectiveness would be between EUR 6000 and 7000 per patient (1€~1.12 USD), based on reduced stay in the ICU and general ward, respectively. Furthermore, predictive use of calprotectin seems to reduce both mortality and the length of hospital stay. This health economic analysis on the predictive use of plasma calprotectin, facilitating clinical decision making in cases of suspected sepsis, indicated that such determination might have a cost-saving and life-saving impact on the healthcare system [66].

Circulating calprotectin levels are significantly correlated with inflammation, trauma severity, and poor outcome at 90 days in patients with severe traumatic brain injury (sTBI). This suggests that circulating calprotectin may be a biomarker to provide additional prognostic information to identify patients at risk of poor outcome after sTBI [107]. Furthermore, in a porcine model of endotoxemic shock that resembles human Gram-negative septic shock, plasma samples were collected before the 6 h endotoxin infusion. There was a significant increase in S-100B during hours 1–5 in comparison with the 0 values, suggesting that endotoxemia causes a small but significant increase in the levels of this widely used brain damage, even if the increase in S-100B could be caused by release from organs other than the brain of possibly blood–brain barrier disruption [108].

There is a frequent antibiotic overuse and subsequent increment in antibiotic resistance. Hence, respiratory tract infections require early diagnosis and adequate treatment in order to distinguish between bacterial and viral infections. In a study comprising 135 patients with bacterial pneumonia, mycoplasma pneumonia, and streptococcal tonsillitis, the diagnostic performance of calprotectin was compared with the performance of heparin-binding protein (HBP) and procalcitonin (PCT). One hundred and forty-four healthy individuals served as controls. Calprotectin was significantly increased in patients with bacterial infections compared with viral infections. PCT was significantly elevated in patients with bacterial pneumonia but not in streptococcal-tonsillitis- or mycoplasma-caused infections. HBP was not able to distinguish between bacterial and viral causes of infections. Rapid detection of bacterial infections may contribute to more selective use of antibiotics [69]. The ability of calprotectin to discriminate between bacterial and viral infections do not seem to limited to an adult population. In 141 febrile infants aged 28–90 days presenting to an emergency department, the usefulness of some biomarkers of infectious diseases (calprotectin, procalcitonin (PCT), C-reactive protein (CRP), and white blood cells (WBCs)) were evaluated. The difference in levels related to antibiotic prescription was significant for all biomarkers but WBCs. The performance of calprotectin in the detection of bacterial infections was comparable to the performance of both PCT and CRP, and superior to the WBC count [109].

## 6. Neutrophil Count

Neutrophils are the most prevalent leukocyte subtype in peripheral blood and serve as frontline defenders against infections and tissue damage. Their antimicrobial functions include phagocytosis, degranulation, production of reactive oxygen species (ROS), and formation of neutrophil extracellular traps (NETs) [110,111,112].

These short-lived granulocytes undergo maturation in the bone marrow over several days, during which they acquire the ability to kill pathogens. Once matured, neutrophils enter circulation and persist for 10–24 h before migrating into tissues, where they remain active for an additional 1–2 days before undergoing apoptosis and removal by phagocytes [113].

Under infectious conditions, the body can shift from steady-state granulopoiesis to emergency granulopoiesis to meet increased neutrophil demand [113]. Their lifespan and function are modulated by cytokines such as IL-1β, IL-2, IL-4, IL-15, IFN-γ, G-CSF, GM-CSF, and lipopolysaccharides, which delay apoptosis and enhance function [114].

Neutrophils are among the first responders at sites of infection or injury. They contribute to microbial killing through NADPH oxidase-mediated ROS generation, degranulation, and NET formation. Additionally, they secrete chemotactic factors that recruit and modulate other immune cells [115]. Initially thought to be activated solely by pathogens, it is now known that endogenous-damage-associated molecular patterns (DAMPs) or alarmins can also trigger neutrophil activation [116].

A key feature of neutrophil activity is the rapid release of pre-formed cytotoxic granules, which contain proteases and microbicidal agents that can be released intracellularly or extracellularly, potentially leading to tissue injury and sustained inflammation [117,118,119,120,121,122,123]. Neutrophil-mediated responses are implicated in the pathogenesis of various chronic conditions, including atherosclerosis, cardiovascular and autoimmune diseases, neurodegeneration, obesity, sepsis, and cancer [123,124,125,126,127,128,129].

Neutrophils also engage in complex crosstalk with other immune and non-immune cells, including platelets, stem cells, and T and B lymphocytes, modulating the inflammatory milieu [124,125,130]. These interactions fine-tune neutrophil survival and activity and influence both innate and adaptive immune responses.

Neutropenia, or reduced neutrophil counts, increases susceptibility to recurrent infections. Conversely, excessive neutrophil accumulation at injury sites may exacerbate inflammation and impair tissue healing [110,131]. Interestingly, neutrophils also contribute to the resolution phase of inflammation by promoting tissue repair and wound healing, although this function is highly context-dependent [132,133,134,135].

In clinical settings, plasma calprotectin levels have been shown to rise as early as two hours after hernia surgery, peaking at 24–36 h postoperatively, suggesting its utility in early detection of surgical inflammation [68]. Similarly, in vitro studies show that calprotectin levels begin increasing within hours of exposure to E. coli or endotoxin [136].

Neutrophil count and the presence of a “left shift” are commonly used, though non-specific, indicators of bacterial infection. Left shift refers to an increased ratio of immature (non-segmented) to mature (segmented) neutrophils in the blood [137]. This phenomenon generally manifests 12–24 h after infection onset and may not be present in the very early phase. Combined analysis of WBC count and left shift offers a reliable reflection of infection dynamics [138].

Not all infections show left shift. In conditions such as bacterial meningitis [139] or abscess formation [140], neutrophil-mediated control may occur without significant bone marrow compensation.

Sepsis introduces profound changes in neutrophil function and distribution. Neutrophils migrate to organs and tissues, reducing circulating levels even amid increased production. NET formation and neutrophil death further contribute to altered counts [141,142]. Accurate interpretation of neutrophil data in sepsis requires understanding of these underlying mechanisms, as typical patterns (e.g., leukocytosis) may be masked.

## 7. C-Reactive Protein (CRP)

CRP was initially identified in patients with *Streptococcus pneumoniae* infection in the laboratory of Oswald Avery [143]. It is a non-specific acute-phase protein composed of five identical ~23 kDa subunits arranged symmetrically around a central pore [144,145]. CRP belongs to the pentraxin family of pattern recognition proteins and plays a critical role in the innate immune response. Synthesized in the liver in response to pro-inflammatory cytokines, CRP enhances complement activation and promotes phagocytosis by macrophages, facilitating the clearance of pathogens [146]. Additionally, it may assist in the removal of necrotic and apoptotic cells [147].

As an acute-phase reactant, CRP production is triggered by cytokines such as IL-1, IL-6, TGF-β, and TNF-α in the setting of tissue injury or inflammation [148,149]. Its serum concentration correlates with the severity of the inflammatory response and is therefore sensitive to even subtle changes [148,150,151]. Due to its short half-life (4–7 h), CRP levels decline rapidly once inflammation resolves, making it a preferred marker for acute conditions [152].

CRP concentrations can increase to 50–100 mg/L within 4–6 h following a mild to moderate inflammatory stimulus, such as cystitis, bronchitis, or uncomplicated skin infections. Levels typically double every 8 h and peak between 36 and 50 h [153]. Mild elevations (2–10 mg/L) are considered indicative of low-grade metabolic inflammation, while levels above 100 mg/L are strongly suggestive of bacterial infection [148].

CRP is useful in monitoring disease progression and treatment response in conditions such as bacterial infections, rheumatoid arthritis, cancer, and acute pancreatitis. Serial CRP measurements may aid in the early detection of neonatal and postoperative sepsis [154,155]. Modest elevations in CRP (3–10 μg/mL) have also been associated with an increased risk of cardiovascular disease, metabolic syndrome, and colorectal cancer [156,157], likely reflecting underlying low-grade chronic inflammation. Other non-inflammatory factors—including genetic polymorphisms, ethnicity, diet, and obesity—can also influence baseline CRP levels [158].

Despite its clinical utility, CRP has limitations. Compared to procalcitonin (PCT), it rises more slowly in response to infection [159], and its low specificity limits its ability to distinguish between infectious and non-infectious inflammation [160,161]. In patients with severe hepatic dysfunction, CRP synthesis may be impaired [151]. Genetic variations affecting IL-6, IL-1, and CRP genes may also influence baseline levels and response [151,162]. While few drugs directly reduce CRP levels, some therapies targeting upstream inflammatory pathways—such as IL-6 inhibitors—can indirectly lower CRP concentrations [151].

## 8. Procalcitonin (PCT)

Procalcitonin (PCT) is a 116-amino acid peptide that serves as the precursor to calcitonin, a hormone produced by thyroid parafollicular (C) cells involved in calcium regulation. Under normal physiological conditions, PCT circulates at low levels (≤0.1 ng/mL) [163]. Alongside CRP, white blood cell count (WBC), and neutrophils, PCT is one of the most widely used biomarkers to differentiate bacterial from viral infections [164].

PCT was first identified as a marker of bacterial infection in 1993, when elevated calcitonin-like immunoreactivity was observed in patients with infections outside the thyroid [165]. In 1994, experimental endotoxin administration to healthy volunteers demonstrated that PCT levels become detectable within 4 h, peak at around 6 h, and plateau at 8–24 h before declining, with a half-life of approximately 24 h [166,167]. Multiple studies have since confirmed PCT’s superior diagnostic accuracy for sepsis and suggest that it may also contribute to the pathophysiological response to systemic infections [168,169]. Clinically, PCT is valuable for assessing infection severity, tracking disease progression, estimating prognosis, and guiding antibiotic therapy.

PCT’s diagnostic utility has been evaluated in various settings, including intensive care units (ICUs), with generally good—but not universally consistent—performance in identifying bacterial infections and sepsis [170]. Notably, a study by Facy et al. reported that CRP was more effective than PCT in detecting postoperative infections [171], and some observational studies in respiratory diseases have shown mixed results. For example, El-Solh et al. found PCT to be inadequate for distinguishing aspiration pneumonia from sterile aspiration pneumonitis [172].

Elevated PCT levels and lack of PCT clearance have been linked to increased mortality in septic patients, with relative risks of 2.60 and 3.05, respectively [173]. A small prospective study by Hamade and Huang found that PCT > 2.0 ng/mL was associated with ICU admission and higher 30-day mortality [174], while another study reported that levels > 0.85 ng/mL predicted *S. pneumoniae* infection [175]. Similar associations with mortality have been observed in studies of sepsis, infective endocarditis, and both community-acquired and ventilator-associated pneumonia [176,177,178,179,180].

Serial PCT measurements may assist in determining appropriate durations for antibiotic therapy, potentially reducing overuse [181,182]. PCT and CRP are the most widely used biomarkers for sepsis, with some suggesting that PCT offers greater specificity [170] and prognostic value [183], although this remains debated [184].

In comparison to PCT, lactic acid levels often show stronger associations with disease severity indices such as the SOFA and APACHE II scores, and with mortality rates [185,186].

Procalcitonin (PCT) levels increase not only in bacterial infections but also in acute malaria—particularly in severe cases—and in fungal infections. However, PCT levels are typically much higher in bacterial sepsis than in these other conditions. In patients with sepsis, a low PCT value may suggest a fungal rather than a bacterial infection [187,188,189,190,191]. Nevertheless, it is important to note that PCT is neither highly sensitive nor specific for diagnosing fungal sepsis. Therefore, PCT results should always be interpreted in the context of the patient’s overall clinical presentation and other laboratory findings.

Additionally, while more specific than CRP, PCT levels can also rise in non-bacterial conditions, such as major trauma, surgery (particularly abdominal), cardiogenic shock, inhalation burns, severe pancreatitis, heatstroke, and rhabdomyolysis [192,193,194,195].

Renal dysfunction can impact PCT interpretation, as patients with chronic kidney disease (CKD) often have higher baseline PCT levels [196]. Furthermore, renal replacement therapies—such as hemodialysis and continuous renal replacement therapy—may lower circulating PCT concentrations [197]. A further limitation is the higher cost of PCT testing compared to CRP, which can restrict its routine use in some clinical settings [197,198].

## 9. Interleukin-6 (IL-6)

Interleukin-6 (IL-6), first described in the mid-1980s by Hirano and colleagues as a T-cell-derived B-cell stimulant [199], is a multifunctional cytokine central to the regulation of inflammation. Alongside IL-1 and TNF, IL-6 is considered one of the most pleiotropic cytokines. It is produced by a wide array of cells in response to infection, tissue injury, inflammation, or malignancy [200,201].

IL-6 gene expression is regulated by several key nuclear factors, including NF-κB [202,203,204], NF-IL6 [205], and HIF-1α [206]. Stimuli that promote IL-6 release include bacterial lipopolysaccharides (LPS) via Toll-like receptors, altered cellular metabolism, and proinflammatory cytokines such as TNF and IL-1 [207,208,209,210,211], as well as viral infections [212].

Elevated IL-6 levels are commonly observed in critical illnesses such as sepsis, acute respiratory distress syndrome (ARDS), and COVID-19, leading to interest in therapeutic strategies aimed at IL-6 inhibition. While IL-6 is primarily proinflammatory, it also exhibits anti-inflammatory and protective roles in certain contexts. It is essential for both innate and adaptive immunity; facilitates pathogen clearance; and regulates diverse physiological processes, including the acute-phase response, hematopoiesis, lipid metabolism, energy balance, and neural development [200].

The IL-6 family includes 10 ligands and 9 receptors, all sharing a common structural core and a signaling component. The IL-6/IL-6 receptor (IL-6R) pathway has been implicated in the progression of diseases such as rheumatoid arthritis, Castleman’s disease, and cytokine release syndrome. Blocking this axis has proven clinically effective in these conditions [213]. Additionally, intracellular signaling via the Janus kinase (JAK) family and STAT proteins contributes to IL-6′s downstream effects. Therapies targeting IL-6, IL-6Rα, or JAK kinases have shown efficacy in multiple immune-mediated diseases [214].

Despite its significance, IL-6 as a biomarker has several limitations. IL-6 levels rise rapidly—within 2 h of endotoxin exposure—but also decline quickly, which may lead to missed diagnoses if sampling is not optimally timed [215]. A further challenge is the absence of standardized cut-off values and clinical interpretation guidelines, which limits its diagnostic consistency across different settings [161]. IL-6 testing is also more expensive than CRP assays, reducing its practicality in routine clinical practice [216].

Additionally, reduced IL-6 production can occur due to genetic factors (e.g., IL-6 gene polymorphisms), specific haplotypes more prevalent in certain populations (such as some Asian groups) [217], IL-6 deficiency [218], or treatment with IL-6 inhibitors [219], which may interfere with diagnostic or prognostic assessments based on IL-6 levels.

## 10. Neutrophil Gelatinase-Associated Lipocalin (NGAL)/Lipocalin-2 (LCN2)

Neutrophil gelatinase-associated lipocalin (NGAL), also known as Lipocalin-2 (LCN2) or human neutrophil lipocalin (HNL), is an acute-phase glycoprotein present in tissues and circulation in three main forms: a 25 kDa monomer; a 45 kDa homodimer; and a 135 kDa heterodimer complexed with matrix metalloproteinase-9 (MMP-9), primarily derived from neutrophil granules. The monomeric 25 kDa form of NGAL is thought to bind small hydrophobic molecules such as bacterial lipopolysaccharides (LPS) and formyl peptides, potentially modulating inflammatory responses. NGAL forms covalent disulfide bonds with MMP-9, a process that may help stabilize extracellular MMP-9.

Although neutrophils are the principal source of NGAL, it is also produced by macrophages and various other cell types, including renal tubular epithelial cells, cardiomyocytes, hepatocytes, pulmonary and gastrointestinal epithelial cells, dendritic cells, and adipocytes. The homodimeric form is specific to neutrophils, with NGAL mRNA expression restricted to immature neutrophils and the protein stored in mature neutrophil granules [220,221,222,223,224,225,226,227,228,229].

NGAL levels have been found to be elevated in the serum of patients with active ulcerative colitis (UC) compared to both healthy controls and patients with Crohn’s disease (CD), suggesting its potential as a biomarker for UC disease activity [230,231]. Furthermore, the MMP-9/NGAL complex has been identified as a surrogate marker of mucosal healing in both UC [222] and CD [232]. In a study by de Bruyn et al., serum MMP-9/NGAL levels were assessed in two independent UC cohorts treated with infliximab. The study found that responders exhibited a significant reduction in MMP-9/NGAL levels, which correlated with mucosal healing and demonstrated a high specificity (91%) in predicting therapeutic response [232].

## 11. Heparin-Binding Protein (HBP)

Heparin-binding protein (HBP), also known as azurocidin or cationic antimicrobial protein of 37 kDa (CAP37), is a granule-derived protein released by activated neutrophils, first identified in 1984 [233]. HBP possesses broad-spectrum antimicrobial activity, targeting a range of pathogens including Gram-positive bacteria (e.g., *Enterococcus faecalis*), Gram-negative bacteria (e.g., *Escherichia coli*), and *Candida albicans*. In addition to its antimicrobial functions, HBP has potent immunomodulatory effects. Similar activities are observed in other neutrophil granule-derived proteins, such as α-defensins and LL-37, collectively referred to as alarmins [234,235].

HBP contributes to the disassembly of the endothelial cytoskeleton, leading to disruption of the vascular barrier, enhanced leukocyte extravasation, and increased vascular permeability [236]. These actions support its key role in host defense during infection. Furthermore, HBP activates monocytes and macrophages, inducing the release of inflammatory mediators such as TNF and interferons (IFNs), which amplify the inflammatory cascade. This amplification has been strongly associated with the development of hypotension and circulatory failure in severe infections [236].

## 12. Neutrophil-Derived Cytokines

Neutrophils are not only effector cells of innate immunity but also serve as sources of a wide array of cytokines. Under both physiological and experimental conditions, neutrophils can produce cytokines from various functional families [237,238,239,240]. The list of neutrophil-derived cytokines and our understanding of their biological roles continue to expand.

For instance, IL-16 has been identified as a preformed cytokine in lysates from resting human neutrophils [241], while cytokines such as CCL23 and IL-23 are produced by human neutrophils activated via Toll-like receptor (TLR) agonists in vitro [242,243]. Supernatants from TLR8-stimulated neutrophils can also promote Th17 polarization in an IL-23-dependent manner [243]. In murine models, CXCR2^+^ neutrophils have been shown to produce IL-5 in the context of resuscitated hemorrhagic shock and tissue trauma [244]. Moreover, murine neutrophils treated with IL-33 in vitro—referred to as N(IL-33) cells—produce Th2 cytokines such as IL-5, IL-4, IL-9, and IL-13 [245].

Adoptive transfer of N(IL-33) neutrophils into mice with ovalbumin-induced allergic asthma resulted in exacerbated lung pathology, likely due to cytokine-mediated effects [245]. Other specialized neutrophil subsets include CD49d^+^ neutrophils, which have been shown to produce CCL5 in response to Sendai virus infection [246] and express CSF-1 in UV-induced skin injury models [247]. Splenic neutrophils have also been reported to secrete IL-36γ in experimental autoimmune encephalomyelitis (EAE) [248], further underscoring the diverse immunoregulatory roles of neutrophils across various disease models.

## 13. Fecal Calprotectin as a Biomarker in Acute Intestinal Infections

Fecal calprotectin is widely recognized as a useful biomarker for distinguishing between bacterial and viral acute intestinal infections. The presence of fecal calprotectin indicates neutrophil infiltration into the intestinal mucosa, which is generally more pronounced in bacterial infections. Median fecal calprotectin values typically fall within the range of 689 to 1870 mg/kg in bacterial infections, while much lower concentrations, often between 89 and 95 mg/kg, are observed in viral cases [249,250]. Using a cutoff of 710 mg/kg, the test demonstrates strong diagnostic performance, with a sensitivity of up to 88.9% and a specificity of 76.0% for identifying bacterial gastroenteritis [249,250]. Studies show that people with bacterial gastroenteritis, such as those caused by Salmonella or Campylobacter, have significantly higher fecal calprotectin levels than people with viral gastroenteritis caused by rotavirus or norovirus. Diagnostic accuracy can be further enhanced by combining fecal calprotectin measurement with other tests, such as fecal occult blood testing [249,250,251].

This biomarker is especially reliable in children over one year old. However, in infants under one year old, naturally higher baseline levels can reduce the test’s specificity [252,253]. Additionally, fecal calprotectin concentrations correlate with infection severity; higher levels reflect more extensive mucosal inflammation and tissue damage in moderate and severe cases [249].

It is important to note that increased fecal calprotectin levels are not exclusive to infections. Elevated levels can also occur in non-infectious conditions, including inflammatory bowel disease [254], nonsteroidal anti-inflammatory drug use [255], and colorectal cancer [254]. This underscores the need to interpret results within the broader clinical context. While fecal calprotectin is effective at indicating the presence of a bacterial infection, it does not provide information about the specific bacterial species involved

## 14. Schematic Presentations of Inflammatory Biomarkers

Table 1 focuses on primary sources, kinetics, half-lives, clinical applications, and diagnostic strengths and limitations for each biomarker, whereas Table 2 compares analytical methods, approximate costs, turnaround times, availability, and clinical impacts for all included markers.

## 15. Summary

Calprotectin is a heterodimeric protein complex consisting of S100A8 and S100A9 subunits that is primarily released by neutrophils and monocytes during inflammation. In addition to its well-established antimicrobial properties, calprotectin plays an important role in immunomodulation, influencing chemotaxis, cytokine secretion, and leukocyte activity. Elevated concentrations of calprotectin in the bloodstream are closely associated with bacterial infections and systemic inflammatory responses, highlighting its potential as a diagnostic marker for conditions such as sepsis and other severe inflammatory diseases. In certain clinical contexts, calprotectin has demonstrated diagnostic performance that is comparable to or even exceeds that of traditional biomarkers such as C-reactive protein (CRP) and procalcitonin (PCT).

## Figures and Tables

**Figure 1 ijms-26-06476-f001:**
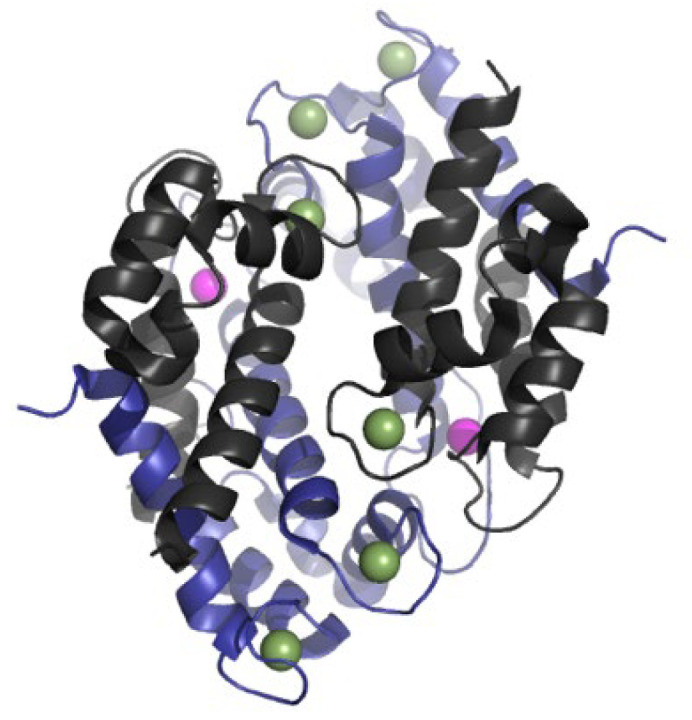
Crystal structure of calprotectin loaded with Mn^2+^ and Ca^2+^, composed of two S100A8-S100A9 dimers. S100A8 and S100A9 chains are shown in grey and blue, respectively. Purple spheres indicate Mn^2+^, and green spheres indicate Ca^2+^. Each dimer binds only one manganese ion [49].

**Figure 2 ijms-26-06476-f002:**
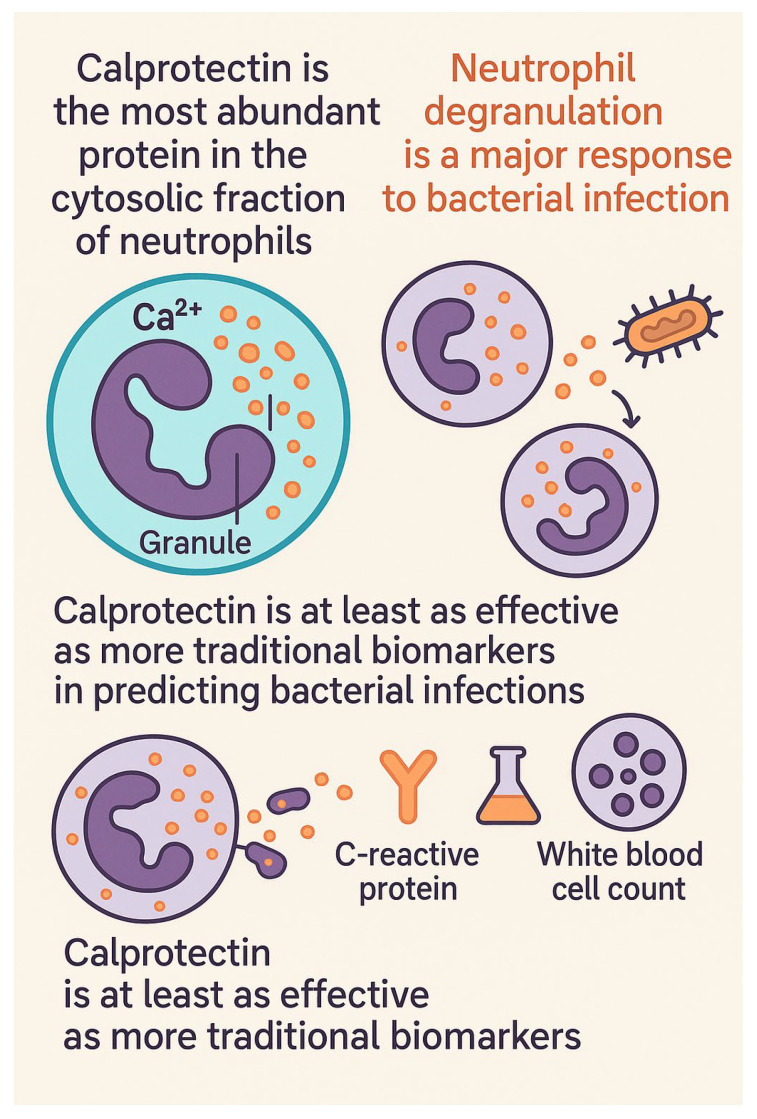
Schematic overview of some characteristics of calprotectin.

**Table 1 ijms-26-06476-t001:** Biological and diagnostic characteristics.

Biomarker	Primary Source	Kinetics (Response Time)	Half-Life	Key Clinical Applications	Diagnostic Strengths	Limitations
WBC count	Bone marrow	4–24 h	6–10 h	General infection screening	Widely available, low cost	Non-specific (stress, steroids)
Neutrophil count	Bone marrow	4–6 h	6–10 h	Acute bacterial infection detection	Rapid, correlates with bacterial burden	Affected by non-infectious inflammation
CRP	Hepatocytes	6–12 h	19 h	General infection/inflammation monitoring	Low cost, widely available	Non-specific, slow rise
PCT	Thyroid/other tissues	3–6 h	24–30 h	Bacterial infection confirmation, sepsis diagnosis, antibiotic stewardship	High specificity for bacterial infections	Cost, delayed rise in localized infections
IL-6	Macrophages, T cells	1–2 h	5 min–15.5 h *	Early sepsis, COVID-19 cytokine storm monitoring	Fastest-rising clinically available cytokine	Short half-life, no standardized cut-offs
NGAL	Neutrophils, epithelia	2–4 h	~10–20 min	Acute kidney injury, bacterial infections	Rapid response to tubular damage	Elevated in CKD, non-infectious inflammation
HBP	Neutrophils	Minutes–hours	~1 h	Sepsis severity, endothelial dysfunction	Predicts organ failure	Limited clinical validation
Circulating calprotectin	Neutrophils, monocytes	2 h	~5 h	Bacterial vs. viral differentiation, sepsis, PJI/UTI diagnosis	Early and sensitive marker, superior for bacterial detection	Elevated in autoimmune/inflammatory conditions
IL-16	Neutrophils	Delayed (necrosis)	Unknown	Inflammation, autoimmunity	Marker of neutrophil death/clearance	Not infection-specific, research use only
CCL23	Neutrophils	Unknown	Unknown	Monocyte/T-cell recruitment	Chemoattractant, immune cell recruitment	Limited clinical validation
IL-23	Dendritic cells, macrophages	Early (hours)	Unknown	Neutrophil activation, bacterial defense	Induces IL-17/IL-22, critical for neutrophil-mediated clearance	Not neutrophil-derived, research use

Abbreviations. CCL23 (chemokine (C-C motif) ligand 23), CRP (C-reactive protein), HBP (heparin binding protein), IL (interleukin), NGAL (neutrophil gelatinase-associated lipocalin), PCT (procalcitonin), and WBC (white blood cell). * IL-6 half-life is context-dependent (minutes in exercise, up to hours in inflammation).

**Table 2 ijms-26-06476-t002:** Practical and economic considerations.

Biomarker	Common Method(s)	Approx. Cost (USD)	Turnaround Time	Routine Availability	Main Clinical Impact
Circulating calprotectin	PETIA/ELISA	USD 15–25	10–90 min	Specialized labs	Rapid PJI/UTI diagnosis *, bacterial–viral differentiation
WBC count	Automated hematology	USD 5–10	15–30 min	All labs	Initial infection screening
Neutrophil count	Automated differential	Included in CBC	15–30 min	All labs	Bacterial infection suspicion
CRP	Immunoturbidimetry	USD 5–10	30–60 min	All labs	General inflammation monitoring
PCT	Chemiluminescent immunoassay	USD 20–40	60–120 min	Major hospitals	Antibiotic stewardship in LRTI/sepsis
IL-6	ELISA	USD 30–50	10–90 min	Major hospitals	Early sepsis triage, COVID-19 severity assessment
NGAL	ELISA/Immunoassay	USD 20–40	1–2 h	Specialized labs	AKI risk stratification
HBP	ELISA	USD 20–40	1–2 h	Research settings	Sepsis severity prediction
IL-16	ELISA	USD 30–50	2–4 h	Research settings	Research marker for inflammation pathways
CCL23	ELISA	USD 30–50	2–4 h	Research settings	Immune cell recruitment studies
IL-23	ELISA	USD 40–60	2–4 h	Research settings	Neutrophil activation research in infections/autoimmunity

Abbreviations are explained in Table 1. Please note that costs and turnaround times are approximate and may vary considerably by method used, region, and laboratory. If all wells on the analysis kit plate should be filled and the plate submitted to a specialized laboratory, the turnaround time may be at least one week. This applies to NGAL, HBP, IL-16, CCL23, and IL-23. * periprosthetic joint infection (PJI) and urinary tract infection (UTI) diagnosis.

## Data Availability

Not applicable.

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
