# Peer review of "Calprotectin as a Biomarker for Infectious Diseases: A Comparative Review with Conventional Inflammatory Markers"

_ijms, 2025, doi:10.3390/ijms26136476_

Round 1

Reviewer 1 Report

Comments and Suggestions for Authors

Dear Authors,

I read the manuscript entitled " Calprotectin as a Biomarker for Infectious Diseases: A Comparative Review with Conventional Inflammatory Markers" by Kristina Sejersen  et al. 

Overall, the paper is well written and quite complete in describing the various inflammatory markers present in infections but also in other pathologies such as pancreatitis, neoplasms, etc. It should be specified that some of these, especially PCR, is not indicated for the diagnosis of fungal infections especially during sepsis. Plasma PCT concentrations, by contrast, increase not only in bacterial infections, but also in acute forms of malaria and fungal infections.

MAjor criticism
the paper suffers from the lack of Tables. In this setting, the authors should build one or more tables in which the authors specific the main targets of the various inflammatory markers, the main differences between one marker and another, the costs of each marker, etc, so that the reader can appreciate the various characteristics of what is described also in order to have a clearer and more detailed vision.

Some references should be added:

  1. Guarino M et al: Presepsin levels and COVID-19 severity: a systematic review and meta-analysis. Clin Exp Med. 2023 Aug;23(4):993-1002. doi: 10.1007/s10238-022-00936-8.
  2. Guarino M et al: Predicting in-hospital mortality for sepsis: a comparison between qSOFA and modified qSOFA in a 2-year single-centre retrospective analysis. Eur J Clin Microbiol Infect Dis. 2021 Apr;40(4):825-831. doi: 10.1007/s10096-020-04086-1

Author Response

R1:

Dear Authors,

I read the manuscript entitled " Calprotectin as a Biomarker for Infectious Diseases: A Comparative Review with Conventional Inflammatory Markers" by Kristina Sejersen  et al.

Overall, the paper is well written and quite complete in describing the various inflammatory markers present in infections but also in other pathologies such as pancreatitis, neoplasms, etc. It should be specified that some of these, especially PCR, is not indicated for the diagnosis of fungal infections especially during sepsis. Plasma PCT concentrations, by contrast, increase not only in bacterial infections, but also in acute forms of malaria and fungal infections.

AUTHORS’ RESPONSE: Thank you for your comment. We assume that your remark referred to procalcitonin (PCT) and not polymerase chain reaction (PCR). As requested, we have added information regarding fungal infections and malaria, with all changes tracked for your review:

“Procalcitonin (PCT) levels increase not only in bacterial infections but also in acute malaria, particularly in severe cases—and in fungal infections. However, PCT levels are typically much higher in bacterial sepsis than in these other conditions. In patients with sepsis, a low PCT value may suggest a fungal rather than a bacterial infection [1-5]. Nevertheless, it is important to note that PCT is neither highly sensitive nor specific for diagnosing fungal sepsis. Therefore, PCT results should always be interpreted in the context of the patient's overall clinical presentation and other laboratory findings”.

References:

1.Li S, Rong H, Guo Q, Chen Y, Zhang G, Yang J. Serum procalcitonin levels distinguish Gram-negative bacterial sepsis from Gram-positive bacterial and fungal sepsis. J Res Med Sci. 2016 Jun 14;21:39. doi: 10.4103/1735-1995.183996. PMID: 27904585; PMCID: PMC5122113.

2.Leli C, Ferranti M, Moretti A, Al Dhahab ZS, Cenci E, Mencacci A. Procalcitonin levels in gram-positive, gram-negative, and fungal bloodstream infections. Dis Markers. 2015;2015:701480. doi: 10.1155/2015/701480. Epub 2015 Mar 17. PMID: 25852221; PMCID: PMC4380090.

3.He S, Ma J, Fan C, Tang C, Chen Y, Xie C. Performance of Procalcitonin to Distinguish Fungal from Bacterial Infections in Patients with Systemic Lupus Erythematosus. Infect Drug Resist. 2021 Nov 16;14:4773-4781. doi: 10.2147/IDR.S337871. PMID: 34815675; PMCID: PMC8605806.

4.Mahittikorn A, Kotepui KU, Mala W, Wilairatana P, Kotepui M. Procalcitonin as a Candidate Biomarker for Malarial Infection and Severe Malaria: A Meta-Analysis. Int J Environ Res Public Health. 2022 Sep 9;19(18):11389. doi: 10.3390/ijerph191811389. PMID: 36141662; PMCID: PMC9517210.

5.Chiwakata CB, Manegold C, Bönicke L, Waase I, Jülch C, Dietrich M. Procalcitonin as a parameter of disease severity and risk of mortality in patients with Plasmodium falciparum malaria. J Infect Dis. 2001 Apr 1;183(7):1161-4. doi: 10.1086/319283. Epub 2001 Mar 1. PMID: 11237849.

MAjor criticism

the paper suffers from the lack of Tables. In this setting, the authors should build one or more tables in which the authors specific the main targets of the various inflammatory markers, the main differences between one marker and another, the costs of each marker, etc, so that the reader can appreciate the various characteristics of what is described also in order to have a clearer and more detailed vision.

AUTHORS’ RESPONSE: Thank you for your valuable feedback regarding the inclusion of comparative tables in our manuscript. We fully agree that such tables enhance clarity and help readers better appreciate the differences among the various inflammatory markers discussed. In response to your suggestion, we have added two comprehensive tables to the revised manuscript. Table 1, titled "Biological and Diagnostic Characteristics of Inflammatory Markers," details the primary sources, kinetics, half-lives, clinical applications, and diagnostic strengths and limitations for each biomarker, including WBC, neutrophil count, CRP, PCT, IL-6, NGAL, HBP, circulating calprotectin, and neutrophil-derived cytokines. Table 2, titled "Practical and Economic Considerations," compares analytical methods, approximate costs, turnaround times, availability, and clinical impacts for all included markers. These tables directly address the request by clearly defining each marker’s biological targets and clinical roles, summarizing their diagnostic performance, and providing essential cost and accessibility information for clinical implementation. We believe these additions offer a more clear and detailed perspective in line with the reviewer recommendations, and we hope they significantly enhance the utility of our paper for both clinicians and researchers.

The Tables are inserted under Section 14.

Some references should be added:

Guarino M et al: Presepsin levels and COVID-19 severity: a systematic review and meta-analysis. Clin Exp Med. 2023 Aug;23(4):993-1002. doi: 10.1007/s10238-022-00936-8.

Guarino M et al: Predicting in-hospital mortality for sepsis: a comparison between qSOFA and modified qSOFA in a 2-year single-centre retrospective analysis. Eur J Clin Microbiol Infect Dis. 2021 Apr;40(4):825-831. doi: 10.1007/s10096-020-04086-1

AUTHORS’ RESPONSE: Thank you very much for your suggestion regarding additional references. We appreciate your input. However, the two references you mentioned—one concerning SOFA and the other about Presepsin—are not directly relevant to our article. We do not compare qSOFA and modified qSOFA in our study, and we do not describe or discuss Presepsin as a biomarker. Therefore, no changes have been made.

However, if the Reviewer finds these references crucial, we are, of course, willing to reconsider and add them to our manuscript. If so, we would appreciate guidance, where they should be inserted.

Reviewer 2 Report

Comments and Suggestions for Authors

Dear authors,

I have read with interest the submitted review, which promotes the use of sophisticated diagnostic techniques for more accurate and timely calprotectin  assessment, such as particle-enhanced turbidimetric immunoassay. Caprotectin is a sensitive biomarker that appear to have promise overall, but more clinical validation and standardization are required to improve its diagnostic usefulness. The review was poor in writing and in presenting the data. However, there are major specific points that should addressed to enhance the quality and precision of the work.

   - The abstract  is too long and has to be reworded.

   - Most of the review titles should be linked with calprotectin and changed as fellow:

Line 134: " Calprotectin Description" should be described in breif and the titled should be modify to " Physicochemical properties and pathophysiological data of Calprotectin"

Line  284: " Neutrophils" should be replaced with " Calprotectin role in neutrophils activation" or " Calprotectin as abiomarker for neutrophils related inflammation"                

     - What is the  Calprotectin's Role in Infection Diagnosis? Please specify its role?

       - What about Fecal Calprotectin as a Biomarker Distinguishing Infectious Cause in Acute Intestinal Infections,

       - The  conclusion and summary need to reworded explained briefly.  

Author Response

R2:

Dear authors,

I have read with interest the submitted review, which promotes the use of sophisticated diagnostic techniques for more accurate and timely calprotectin  assessment, such as particle-enhanced turbidimetric immunoassay. Caprotectin is a sensitive biomarker that appear to have promise overall, but more clinical validation and standardization are required to improve its diagnostic usefulness. The review was poor in writing and in presenting the data. However, there are major specific points that should addressed to enhance the quality and precision of the work.

   - The abstract  is too long and has to be reworded.

AUTHORS’ RESPONSE: Thank you for your comment regarding the abstract length and wording. The abstract has been revised and shortened as suggested, with all changes highlighted using track changes in the revised manuscript:

“Calprotectin, the most abundant cytosolic protein in neutrophils, is a S100A8/S100A9 heterodimer released during immune activation. It inhibits bacterial growth by binding to essential metal ions and contributes to inflammation and leukocyte migration. This review highlights calprotectin's potential as a diagnostic marker for bacterial infections and inflammation. Clinical trials demonstrate that calprotectin is at least as effective as C-reactive protein, procalcitonin, and white blood cell counts in predicting bacterial infections. The rapid elevation of calprotectin levels in the early stages of sepsis, pneumonia, brain injury, and transplant complications underscores its diagnostic value. Predictive use of calprotectin may reduce ICU stays, mortality, and costs. However, challenges remain, including assay standardization and bacterial-viral differentiation. Advanced methods, such as the particle-enhanced turbidimetric immunoassay, enable faster and more reliable measurements. While calprotectin shows promise, further standardization and clinical validation are necessary to optimize its diagnostic utility”.

   - Most of the review titles should be linked with calprotectin and changed as fellow:

Line 134: " Calprotectin Description" should be described in breif and the titled should be modify to " Physicochemical properties and pathophysiological data of Calprotectin"

AUTHORS’ RESPONSE: Thank you for your comment. The changes have been made and are visible with "track changes."

Line  284: " Neutrophils" should be replaced with " Calprotectin role in neutrophils activation" or " Calprotectin as abiomarker for neutrophils related inflammation"

AUTHORS’ RESPONSE: Thank you for your comment. We would like to clarify that this section refers to the discussion of neutrophils as a separate biomarker per se for inflammation and infection, and is in this context not related to calprotectin. To clarify this and to avoid misunderstandings, we have changed "neutrophils" to "neutrophil count" in the text. These changes have been implemented and are tracked using "track changes."

     - What is the  Calprotectin's Role in Infection Diagnosis? Please specify its role?

AUTHORS’ RESPONSE: Thank you for your valuable comment. As requested, we have clarified and specified the role of calprotectin in infection diagnosis in the revised manuscript. These changes have been implemented and are tracked using "track changes".

The following para is inserted under section 4.1.

Circulating calprotectin, which is detectable in plasma or serum, is a valuable biomarker for diagnosing infections. It is particularly useful for identifying and evaluating the severity of bacterial infections and sepsis. It is released promptly into the bloodstream following neutrophil activation during infection, enabling early identification of the body’s inflammatory response, often before notable increases in conventional markers like C-reactive protein (CRP) and procalcitonin (PCT) occur [1-3]. Recent research indicates that calprotectin levels can rise within two hours of inflammation onset [4].

Circulating calprotectin concentrations are markedly higher in bacterial infections than in viral or non-infectious conditions. This makes calprotectin a helpful marker for distinguishing bacterial from viral causes, especially in cases of acute respiratory infections and sepsis. Multiple studies have shown that calprotectin surpasses or complements other biomarkers, such as PCT and HBP, in distinguishing between bacterial and viral infections. Furthermore, evidence suggests that calprotectin is more effective than PCT and HBP in distinguishing bacterial, mycoplasma, and viral infections.

Beyond its diagnostic role, circulating calprotectin is closely associated with infection severity and prognosis. Elevated levels are commonly observed in patients with sepsis and multi-organ dysfunction. Higher concentrations at admission are associated with an increased risk of adverse outcomes, including 30-day mortality in critically ill and septic patients [1-3, 5, 6]. The relationship between calprotectin levels and infection severity highlights its usefulness in risk assessment and guiding clinical decisions, particularly in emergency and intensive care settings.

Furthermore, circulating calprotectin has shown utility in the context of SARS-CoV-2 infection, with significantly higher levels observed in patients with confirmed infection compared to controls, supporting its role in early diagnosis and patient management [7].

However, calprotectin is not exclusively specific to infections; it is also a sensitive indicator of neutrophil-mediated inflammation and may be elevated in other inflammatory disorders, such as autoimmune diseases. Therefore, its levels should always be interpreted within the broader clinical context [8].

Calprotectin is increasingly recognized as a valuable marker of infection and inflammation in various clinical specimens, including body fluids such as synovial fluid, cerebrospinal fluid (CSF), and pleural fluid, beyond blood and stool.

Urine calprotectin has shown promise as a useful biomarker for detecting urinary tract infections (UTIs). Studies have shown that people with UTIs have significantly higher levels of urinary calprotectin. Its diagnostic performance is comparable to or better than conventional indicators, such as dipstick pyuria. Importantly, calprotectin can detect infections when pyuria is absent, which could improve diagnostic sensitivity in atypical cases [9].

Calprotectin in synovial fluid is a highly sensitive and specific biomarker for diagnosing periprosthetic joint infection (PJI). Meta-analyses and clinical studies report sensitivity and specificity rates exceeding 90%, establishing synovial calprotectin as a reliable, rapid diagnostic tool for distinguishing septic from aseptic joint failure, even in complex clinical scenarios. Calprotectin can be measured using ELISA or lateral flow assays. Its low cost and high diagnostic accuracy support its use in routine clinical practice [10-12].

Calprotectin concentrations in CSF are notably increased in individuals with infections or inflammatory disorders of the central nervous system. Research indicates that measuring CSF calprotectin levels is useful for distinguishing bacterial meningitis from non-bacterial forms, demonstrating strong sensitivity and specificity. Elevated calprotectin levels also signal neuroinflammatory processes and are significantly higher in patients with neurological infections than in healthy controls. However, while calprotectin is a reliable indicator of inflammation, it does not always allow for clear differentiation between infectious and non-infectious inflammatory conditions within the CNS [13,14].

Calprotectin concentrations in pleural fluid are significantly higher in bacterial infections, such as parapneumonic effusions and empyema, than in malignant pleural effusions. Research indicates that benign pleural effusions, particularly those caused by infections such as pneumonia or tuberculosis, often exhibit calprotectin levels in the thousands of ng/mL. For instance, one study reported median calprotectin concentrations of 3,517.9 ng/mL in effusions related to pneumonia and 2,982.3 ng/mL in effusions due to tuberculosis. In contrast, malignant effusions had a median concentration of approximately 257.2 ng/mL [15].

Measuring calprotectin in synovial fluid, CSF, and pleural fluid provides valuable clinical information for diagnosing infections and inflammatory conditions in these compartments. Calprotectin's high sensitivity and specificity, rapid turnaround time, and ease of measurement make it a promising tool for clinicians in diverse diagnostic settings.

References

  1. Jonsson N, Nilsen T, Gille-Johnson P, Bell M, Martling CR, Larsson A, Mårtensson J. Calprotectin as an early biomarker of bacterial infections in critically ill patients: an exploratory cohort assessment. Crit Care Resusc. 2017 Sep;19(3):205-213. PMID: 28866970.
  2. Havelka A, Larsson AO, Mårtensson J, Bell M, Hultström M, Lipcsey M, Eriksson M. Analysis of Calprotectin as an Early Marker of Infections Is Economically Advantageous in Intensive Care-Treated Patients. Biomedicines. 2023 Aug 1;11(8):2156. doi: 10.3390/biomedicines11082156. PMID: 37626653; PMCID: PMC10452832.
  3. Larsson A, Tydén J, Johansson J, Lipcsey M, Bergquist M, Kultima K, Mandic-Havelka A. Calprotectin is superior to procalcitonin as a sepsis marker and predictor of 30-day mortality in intensive care patients. Scand J Clin Lab Invest. 2020 Feb-Apr;80(2):156-161. doi: 10.1080/00365513.2019.1703216. Epub 2019 Dec 14. PMID: 31841042.
  4. Sejersen K, Havelka A, Sanchez Salas P, Larsson A. Early kinetics of calprotectin in plasma following inguinal hernia surgery. Innate Immun. 2022 Jan;28(1):49-54. doi: 10.1177/17534259211069635. Epub 2022 Feb 1. PMID: 35102763; PMCID: PMC8841635.
  5. Havelka A, Sejersen K, Venge P, Pauksens K, Larsson A. Calprotectin, a new biomarker for diagnosis of acute respiratory infections. Sci Rep. 2020 Mar 6;10(1):4208. doi: 10.1038/s41598-020-61094-z. PMID: 32144345; PMCID: PMC7060262.
  6. Wirtz TH, Buendgens L, Weiskirchen R, Loosen SH, Haehnsen N, Puengel T, Abu Jhaisha S, Brozat JF, Hohlstein P, Koek G, Eisert A, Mohr R, Roderburg C, Luedde T, Trautwein C, Tacke F, Koch A. Association of Serum Calprotectin Concentrations with Mortality in Critically Ill and Septic Patients. Diagnostics (Basel). 2020 Nov 23;10(11):990. doi: 10.3390/diagnostics10110990. PMID: 33238644; PMCID: PMC7700375.
  7. Silvin A, Chapuis N, Dunsmore G, Goubet AG, Dubuisson A, Derosa L, Almire C, Hénon C, Kosmider O, Droin N, Rameau P, Catelain C, Alfaro A, Dussiau C, Friedrich C, Sourdeau E, Marin N, Szwebel TA, Cantin D, Mouthon L, Borderie D, Deloger M, Bredel D, Mouraud S, Drubay D, Andrieu M, Lhonneur AS, Saada V, Stoclin A, Willekens C, Pommeret F, Griscelli F, Ng LG, Zhang Z, Bost P, Amit I, Barlesi F, Marabelle A, Pène F, Gachot B, André F, Zitvogel L, Ginhoux F, Fontenay M, Solary E. Elevated Calprotectin and Abnormal Myeloid Cell Subsets Discriminate Severe from Mild COVID-19. Cell. 2020 Sep 17;182(6):1401-1418.e18. doi: 10.1016/j.cell.2020.08.002. Epub 2020 Aug 5. PMID: 32810439; PMCID: PMC7405878.
  8. Carnazzo V, Redi S, Basile V, Natali P, Gulli F, Equitani F, Marino M, Basile U. Calprotectin: two sides of the same coin. Rheumatology (Oxford). 2024 Jan 4;63(1):26-33. doi: 10.1093/rheumatology/kead405. PMID: 37603715; PMCID: PMC10765140.
  9. Waldecker-Gall S, Waldecker CB, Babel N, Baraliakos X, Seibert F, Westhoff TH. Urinary calprotectin as a diagnostic tool for detecting significant bacteriuria. Sci Rep. 2024 May 28;14(1):12230. doi: 10.1038/s41598-024-62605-y. PMID: 38806578; PMCID: PMC11133377.
  10. Hantouly AT, Salameh M, Toubasi AA, Salman LA, Alzobi O, Ahmed AF, Hameed S, Zikria B, Ahmed G. Synovial fluid calprotectin in diagnosing periprosthetic joint infection: A meta-analysis. Int Orthop. 2022 May;46(5):971-981. doi: 10.1007/s00264-022-05357-6. Epub 2022 Mar 2. PMID: 35233711; PMCID: PMC9001224.
  11. Suren C, Lazic I, Haller B, Pohlig F, von Eisenhart-Rothe R, Prodinger P. The synovial fluid calprotectin lateral flow test for the diagnosis of chronic prosthetic joint infection in failed primary and revision total hip and knee arthroplasty. Int Orthop. 2023 Apr;47(4):929-944. doi: 10.1007/s00264-023-05691-3. Epub 2023 Jan 19. PMID: 36656361; PMCID: PMC10014771.
  12. Lazic, I., Burdach, A., Pohlig, F. et al. Utility of synovial calprotectin lateral flow test to exclude chronic prosthetic joint infection in periprosthetic fractures: a prospective cohort study. Sci Rep 12, 18385 (2022). https://doi.org/10.1038/s41598-022-22892-9
  13. Lin Q, Huang E, Fan K, Zhang Z, Shangguan H, Zhang W, Fang W, Ou Q, Liu X. Cerebrospinal Fluid Neutrophil Gelatinase-Associated Lipocalin as a Novel Biomarker for Postneurosurgical Bacterial Meningitis: A Prospective Observational Cohort Study. Neurosurgery. 2024 Dec 1;95(6):1418-1428. doi: 10.1227/neu.0000000000003021. Epub 2024 Jun 10. PMID: 38856216.
  14. Dastych M, Gottwaldová J, Čermáková Z. Calprotectin and lactoferrin in the cerebrospinal fluid; biomarkers utilisable for differential diagnostics of bacterial and aseptic meningitis? Clin Chem Lab Med. 2015 Mar;53(4):599-603. doi: 10.1515/cclm-2014-0775. PMID: 25405719.
  15. Sánchez-Otero N, Blanco-Prieto S, Páez de la Cadena M, Vázquez-Iglesias L, Fernández-Villar A, Botana-Rial MI, Rodríguez-Berrocal FJ. Calprotectin: a novel biomarker for the diagnosis of pleural effusion. Br J Cancer. 2012 Nov 20;107(11):1876-82. doi: 10.1038/bjc.2012.478. Epub 2012 Oct 23. PMID: 23093228; PMCID: PMC3504943.

       - What about Fecal Calprotectin as a Biomarker Distinguishing Infectious Cause in Acute Intestinal Infections,

AUTHORS’ RESPONSE: Thank you for your comment. As requested, we have added information regarding fecal calprotectin, with all changes tracked for your review:

“Fecal calprotectin is widely recognized as a useful biomarker for distinguishing between bacterial and viral acute intestinal infections. The presence of fecal calprotectin indicates neutrophil infiltration into the intestinal mucosa, which is generally more pronounced in bacterial infections. Median fecal calprotectin values typically fall within the range of 689 to 1,870 mg/kg in bacterial infections, while much lower concentrations, often between 89 and 95 mg/kg, are observed in viral cases [1,2]. Using a cutoff of 710 mg/kg, the test demonstrates strong diagnostic performance, with a sensitivity of up to 88.9% and a specificity of 76.0% for identifying bacterial gastroenteritis [1,2]. Studies show that people with bacterial gastroenteritis, such as those caused by Salmonella or Campylobacter, have significantly higher fecal calprotectin levels than people with viral gastroenteritis caused by rotavirus or norovirus. Diagnostic accuracy can be further enhanced by combining fecal calprotectin measurement with other tests, such as fecal occult blood testing [1-3].

This biomarker is especially reliable in children over one year old. However, in infants under one year old, naturally higher baseline levels can reduce the test's specificity [4,5]. Additionally, fecal calprotectin concentrations correlate with infection severity; higher levels reflect more extensive mucosal inflammation and tissue damage in moderate and severe cases [1].

It is important to note that increased fecal calprotectin levels are not exclusive to infections. Elevated levels can also occur in non-infectious conditions, including inflammatory bowel disease [6], nonsteroidal anti-inflammatory drug use [7], and colorectal cancer [6]. This underscores the need to interpret results within the broader clinical context. While fecal calprotectin is effective at indicating the presence of a bacterial infection, it does not provide information about the specific bacterial species involved”.

References

  1. Chen CC, Huang JL, Chang CJ, Kong MS. Fecal calprotectin as a correlative marker in clinical severity of infectious diarrhea and usefulness in evaluating bacterial or viral pathogens in children. J Pediatr Gastroenterol Nutr. 2012 Nov;55(5):541-7. doi: 10.1097/MPG.0b013e318262a718. PMID: 22699836.
  2. Duman M, Gencpinar P, Biçmen M, Arslan N, Özden Ö, Üzüm Ö, Çelik D, Sayıner AA, Gülay Z. Fecal calprotectin: can be used to distinguish between bacterial and viral gastroenteritis in children? Am J Emerg Med. 2015 Oct;33(10):1436-9. doi: 10.1016/j.ajem.2015.07.007. Epub 2015 Jul 6. PMID: 26233616.
  3. Lué A, Hijos G, Sostres C, Perales A, Navarro M, Barra MV, Mascialino B, Andalucia C, Puente JJ, Lanas Á, Gomollon F. The combination of quantitative faecal occult blood test and faecal calprotectin is a cost-effective strategy to avoid colonoscopies in symptomatic patients without relevant pathology. Therap Adv Gastroenterol. 2020 May 18;13:1756284820920786. doi: 10.1177/1756284820920786. PMID: 32523623; PMCID: PMC7235671.
  4. Zeng J, Yu W, Gao X, Yu Y, Zhou Y, Pan X. Establishing reference values for age-related fecal calprotectin in healthy children aged 0-4 years: a systematic review and meta-analysis. 2025 Jun 12;13:e19572. doi: 10.7717/peerj.19572. PMID: 40525108; PMCID: PMC12169166.
  5. Roca, M., Rodriguez Varela, A., Carvajal, E. et al.Fecal calprotectin in healthy children aged 4–16 years. Sci Rep 10, 20565 (2020). https://doi.org/10.1038/s41598-020-77625-7
  6. Røseth AG, Fagerhol MK, Aadland E, Schjønsby H. Assessment of the neutrophil dominating protein calprotectin in feces. A methodologic study. Scand J Gastroenterol. 1992 Sep;27(9):793-8. doi: 10.3109/00365529209011186. PMID: 1411288.
  7. Bjarnason I, MacPherson A, Hollander D. Intestinal permeability: an overview. 1995 May;108(5):1566-81. doi: 10.1016/0016-5085(95)90708-4. PMID: 7729650.

       - The  conclusion and summary need to reworded explained briefly.

AUTHORS’ RESPONSE: Thank you for your comment. As requested, we reworded summary with all changes tracked for your review:

“Calprotectin is a heterodimeric protein complex consisting of S100A8 and S100A9 subunits that is primarily released by neutrophils and monocytes during inflammation. In addition to its well-established antimicrobial properties, calprotectin plays an important role in immunomodulation, influencing chemotaxis, cytokine secretion and leukocyte activity. Elevated concentrations of calprotectin in the bloodstream are closely associated with bacterial infections and systemic inflammatory responses, highlighting its potential as a diagnostic marker for conditions such as sepsis and other severe inflammatory diseases. In certain clinical contexts, calprotectin has demonstrated diagnostic performance that is comparable to or even exceeds that of traditional biomarkers such as C-reactive protein (CRP) and procalcitonin (PCT)”.

Round 2

Reviewer 2 Report

Comments and Suggestions for Authors

Dear authors,

Thank you for your response!